# Role of a Novel Heparanase Inhibitor on the Balance between Apoptosis and Autophagy in U87 Human Glioblastoma Cells

**DOI:** 10.3390/cells12141891

**Published:** 2023-07-19

**Authors:** Valeria Manganelli, Roberta Misasi, Gloria Riitano, Antonella Capozzi, Vincenzo Mattei, Tuba Rana Caglar, Davide Ialongo, Valentina Noemi Madia, Antonella Messore, Roberta Costi, Roberto Di Santo, Maurizio Sorice, Tina Garofalo

**Affiliations:** 1Department of Experimental Medicine, “Sapienza” University, 00161 Rome, Italy; 2Biomedicine and Advanced Technologies Rieti Center, Sabina Universitas, 02100 Rieti, Italy; 3Dipartimento di Chimica e Tecnologie del Farmaco, Istituto Pasteur-Fondazione Cenci Bolognetti, “Sapienza” University of Rome, 00185 Rome, Italy

**Keywords:** heparanase inhibitor, apoptosis, autophagy, U87 human glioblastoma cells

## Abstract

Background: Heparanase (HPSE) is an endo-β-glucuronidase that cleaves heparan sulfate side chains, leading to the disassembly of the extracellular matrix, facilitating cell invasion and metastasis dissemination. In this research, we investigated the role of a new HPSE inhibitor, RDS 3337, in the regulation of the autophagic process and the balance between apoptosis and autophagy in U87 glioblastoma cells. Methods: After treatment with RDS 3337, cell lysates were analyzed for autophagy and apoptosis-related proteins by Western blot. Results: We observed, firstly, that LC3II expression increased in U87 cells incubated with RDS 3337, together with a significant increase of p62/SQSTM1 levels, indicating that RDS 3337 could act through the inhibition of autophagic-lysosomal flux of LC3-II, thereby leading to accumulation of lipidated LC3-II form. Conversely, the suppression of autophagic flux could activate apoptosis mechanisms, as revealed by the activation of caspase 3, the increased level of cleaved Parp1, and DNA fragmentation. Conclusions: These findings support the notion that HPSE promotes autophagy, providing evidence that RDS 3337 blocks autophagic flux. It indicates a role for HPSE inhibitors in the balance between apoptosis and autophagy in U87 human glioblastoma cells, suggesting a potential role for this new class of compounds in the control of tumor growth progression.

## 1. Introduction

The endo-β-glucuronidase heparanase (HPSE) is the primary enzyme that cleaves heparan sulfate (HS) side chains linked to proteoglycan core proteins, leading to the disassembly of the extracellular matrix (ECM). This event is therefore involved in the main biological phenomena associated with tissue remodeling and cell invasion, including inflammation, angiogenesis, and metastasis [1,2,3,4].

HPSE is synthesized in the endoplasmic reticulum as a precursor of 68 kDa, which, in the Golgi, is then processed in proHPSE (65 kDa). After secretion in the extracellular space as a latent 65-kDa enzyme, HPSE rapidly interacts with membrane heparan sulfate proteoglycans (HSPGs), such as syndecans [5,6] for being endocytosed and processed into a highly active 50-kDa enzyme [7]. HPSE has been shown to reside primarily within endocytic vesicles, and it assumes a polar, perinuclear localization colocalizing with lysosomal markers [8]. Recent studies have identified cathepsin L, a lysosomal protease, responsible for the processing and activation of HPSE [9,10]. Activated HPSE may have different destinations in the cell; it can be secreted, included in autophagosomes, shuttled into the nucleus, or anchored on the surface of exosomes [11,12,13].

Previous studies illustrated that HPSE expression is increased in different types of tumors (i.e., carcinomas and sarcomas) and hematologic malignancies [3,14,15,16].

The role of HPSE in the development of cancers is mainly due to its degradation activity of HS, which facilitates metastasis dissemination and cell invasion. HPSE has been proven to be involved in the autophagy process since it is kept in lysosomes following secretion [12]. Several studies provide evidence that enhanced tumor growth and chemoresistance exerted by HPSE are mediated, in part, by increasing autophagy. A role for HPSE inside the cell has been described, especially regarding its non-enzymatic activity that plays an essential role in regulating a variety of biological functions [17,18].

Autophagy is a homeostatic mechanism through which cytoplasmic contents, including macromolecules or organelles, are sequestered into autophagosomes, which are double-membrane vesicles. After their fusion with lysosomes, the intracellular material is degraded and recycled [19,20,21,22].

In the setting of cancer, the function of the autophagic process is complex. Whereas some studies prove that autophagy can be beneficial for tumor cell survival [23,24], others indicate that the self-degradative process may prevent tumor growth during the early stages of tumorigenesis [25,26].

Since the elevation of HPSE expression in cancer is often associated with more aggressive disease and poor prognosis, this has encouraged the development of HPSE inhibitors [27,28].

HPSE can be defined as a multifunctional protein that is involved in the establishment and development of numerous diseases. Once inactivated, there are no other enzymes capable of performing the same function. It is evident that HPSE can be an effective and promising therapy target.

Several HPSE inhibitors were identified in the last two decades that were able to neutralize extracellular HPSE and decrease its intracellular contents [27,29]. These range from monoclonal antibodies, heparin derivatives and polysulfated oligosaccharides, nucleic acids, proteins, and small molecules. However, despite the intensive efforts in this field of research, no compounds capable of inhibiting or modulating HPSE activity have reached the clinical stage, while only four drugs have entered the clinical trials as anticancer agents. These polysulfated polysaccharides show several limits, mainly related to their heterogeneous nature and high molecular weight, thus hindering the characterization, standardization, evaluation of biological data, and patient compliance, due to their parental administration route. Differently, the development of small molecule compounds is more encouraging, given that they can overcome some limits of polysaccharides. Indeed, small molecules, owing to their small size, are highly advantageous thanks to their ability to hit both extracellular and intracellular targets—as they can easily cross the cellular plasma membrane—their more suitable pharmacokinetic characteristics, and the capability of being optimized for oral administration. Moreover, small molecules can be easily and cheaply synthesized, and most of the drugs generated by pharmaceutical companies are still small molecules [30]. Previously, several asymmetrical and symmetrical small molecule derivatives characterized by a benzazolyl-5-acetate scaffold were synthesized and described as potent HPSE inhibitors endowed with nanomolar/micromolar potencies [31,32]. Among them, the symmetrical benzazole compounds were found more effective in inhibiting HPSE with respect to the compounds characterized by an asymmetrical shape.

In particular, in this study, we used the symmetrical inhibitor RDS 3337 (Figure 1), characterized by a benzoxazol-5-yl acetic acid scaffold functionalized with a central thiourea moiety and a glycine residue, and endowed with high anti-HPSE activity in the nanomolar range (80 nM). It is worthy of note that, while RDS 3337 is less potent than roneparstat, one of the most potent inhibitors of HPSE described (that is, in clinical trials), the amount potentially needed for clinical use could be about half of that of roneparstat, due to their enormous difference in molecular weight. Of interest, it has been shown that this compound was able to reduce the in vitro invasive ability of human glioblastoma cell lines (U87), although the underlying mechanisms involved are not completely clear. In this research, we investigated the role of the HPSE inhibitor RDS 3337 (7 g) [33] in the regulation of the autophagic process and the balance between apoptosis and autophagy in U87 human glioblastoma cells.

## 2. Materials and Methods

### 2.1. Cell Cultures and Treatments

RPE-1 human non-cancer neuro-ectodermal cells (ATCC) were grown in DMEM-F12, and human SK-N-BE2 (ATCC) neuroblastoma cells were grown in an RPM1 1640 medium. Human glioblastoma U87 cells (ATCC, Manassas, VA, USA) were grown in Dulbecco’s modified Eagle medium (DMEM, Sigma-Aldrich, St. Louis, MO, USA). All culture media were supplemented with 10% fetal bovine serum (FBS) at 37 °C in a humified 5% CO_2_ atmosphere. All experiments were carried out using cells split no more than seven times. For autophagy induction, RPE1 cells were stimulated under the condition of nutrient deprivation with Hanks’ Balanced Salt Solution (HBSS, Sigma H9269) for 16 h at 37 °C. The optimal incubation time with HBSS was selected on the basis of preliminary experiments. To inhibit autophagic flux, 100 nM Bafilomycin A1 (Baf A1) (Sigma-Aldrich B1793) was added 2 h before lysis.

### 2.2. In Vitro Treatment with HPSE Inhibitors

Cells were treated with the benzazolyl derivative RDS 3337 endowed with potent anti-HPSE activity, as described previously [33]. The compound RDS 3337 was dissolved in dimethyl sulfoxide (DMSO, Sigma-Aldrich) at the 10 mM stock solution and then used at the concentration of 80, 320, and 1280 nM. We considered vehicle cells without any treatment, with only culture medium plus DMSO used as a vehicle to dissolve the compound. HPSE activity was also evaluated, as reported in Appendix A.

### 2.3. Cell Viability Assays

#### 2.3.1. Trypan Blue Assay

As previously reported [33], Trypan Blue (Sigma-Aldrich) assay was used to evaluate the cell viability of both RPE-1 and U87 cells. Cells were seeded into cell culture plates at a concentration of 5 × 10^5^ cells/mL and kept for 24 h at 37 °C with 5% CO_2_. Then, cells were treated with different concentrations of RDS 3337 (80, 320, 1280 nM) for an incubation time of 24, 48, or 72 h. Vehicle-treated cells or cells incubated with RDS 3337 were analyzed by Trypan Blue (Sigma-Aldrich) assay to assess cell viability. DMSO is the vehicle to dissolve the compound, and we consider cells with only DMSO as vehicle-treated cells. All experiments were carried out in quintuplicate.

#### 2.3.2. WST-1 Cell Viability Assay

The viability of RPE-1 and U87 cells, incubated with the benzazolyl derivative RDS 3337, was assessed by performing spectrometric quantification with 2-(4-iodophenyl)-3-(4-nitrophenyl)-5-(2,4-disulfophenyl)-2h-tetrazolium and monosodium salt (WST-1) reagent (Abcam, 65473, Cambridge, UK), in accord with manufacturer’s instructions. Cells were seeded into 96-well cell culture plates at a concentration of 10^4^ cells/mL in a final volume of 100 μL culture medium, and then treated with different concentrations of RDS 3337 (80, 320, 1280 nM) for incubation times of 24, 48, or 72 h at 37 °C with 5% CO_2_. Then, 10 μL of WST-1 was added to the culture wells. After that, cells were incubated at 37 °C for 2 h. The absorbance was measured at 450 nm through the GloMax^®^-Multi Detection System (Promega, Madison, WI, USA).

#### 2.3.3. BrdU Cell Proliferation Assay

Cell proliferation was analyzed by the BrdU/anti-BrdU method. Cells were seeded in 6-well plates (5 × 10^5^) in the presence or absence of the test compound and incubated for 24, 48, or 72 h. Next, cells were labeled for 90 min with 10 µM of BrdU (Sigma-Aldrich) and washed twice with ice-cold PBS. Subsequently, cells were detached, and the cell pellet was resuspended in ice-cold PBS and fixed in ice-cold acetone/methanol (1:5, *v*:*v*) for 1 h at 4 °C. Cells were washed twice with PBS containing Tween 20 (0.5%), followed by incubation with 2 N HCl (Merck, Darmstadt, Germany) for 45 min at room temperature. The cells were then washed twice, and the pellet resuspended in 0.1 M Na_2_B_4_O_7_; after washing in PBS/Tween, cells were incubated with FITC-conjugated anti-BrdU (Becton Dickinson Biosciences, San Jose, CA, USA) at RT for 30 min. After two washes in PBS/Tween, the cells were analyzed by a CytoFlex flow cytometer (Beckman Coulter, Coulter Brea, CA, USA).

### 2.4. Preparation of Cell Extracts

Cells untreated or treated with RDS 3337 for 18 or 72 h at 37 °C in 5% CO_2_ in the presence or absence of bafilomycin A1 (Baf A1; 100 nM) were lysed in lysis buffer, containing 20 mM HEPES, pH 7.2; 1% Nonidet P-40, 10% glycerol, 50 mM NaF, 1 mM Na_3_VO_4_, and a protease inhibitors cocktail (Sigma-Aldrich). Proteins were recovered after centrifugation of lysates at 15,000× *g* for 15 min at 4 °C. Then, samples were resuspended in a lysis buffer, and whole-cell extracts were obtained, as reported above. Protein contents were determined by Bradford assay.

### 2.5. Western Blot Analysis for Autophagy and Apoptosis-Related Proteins

Proteins were separated by polyacrylamide gel electrophoresis (15% or 7.5%) with sodium dodecyl sulfate (SDS-PAGE) and transferred onto polyvinylidene fluoride (PVDF) membranes (Bio-Rad, Hercules, CA, USA). Membranes were blocked by 5% milk in TBS-Tween (Tris-Buffered Saline 0.05% Tween 20) for 1 h, and then incubated with rabbit anti-LC3 pAb (NB100-2331 Novus Biologicals, Centennial, CO, USA), rabbit anti-SQSTM1 mAb (8025, Cell Signaling Technology, Boston, MA, USA), rabbit anti-PARP monoclonal antibody mAb (9542, Cell Signaling Technology), or rabbit anti-caspase 3 mAb (9661, Cell Signaling Technology), all of them diluted 1:1000 in TBS-Tween. Membranes were incubated with horseradish peroxidase (HRP)-conjugated anti-rabbit IgG (A1949, Sigma-Aldrich). Mouse anti-β-actin mAb (A5316, Sigma-Aldrich) and horseradish peroxidase (HRP)-conjugated anti-mouse IgG antibody (NA931V, Sigma-Aldrich) were used to have a loading control. Immunoreactivity was assessed through the development of a chemiluminescence reaction using an ECL detection system (Amersham, Buckinghamshire, UK). Densitometric analysis was performed by Mac OS X (Apple Computer International, Cupertino, CA, USA), using NIH Image 1.62 software.

### 2.6. Autophagy Evaluation by Flow Cytometry

RPE-1 cells were starved with HBSS for 16 h, or treated with 320 nM RDS 3337 for 72 h in the presence or absence of bafilomycin A1 (Baf A1; 100 nM). At the end of treatment, cells were analyzed by flow cytometry after single staining with a Cyto-ID detection kit (ENZ-51031-K200, Enzo Life Sciences, Exeter, UK). This assay was optimized for the evaluation of autophagy at the cellular level by flow cytometry using a 488 nm-excitable probe that becomes fluorescent in autophagic vesicles (autophagosomes) produced during autophagy. To detect p62/SQSTM1 levels, cells were analyzed by flow cytometry after fixation with 4% paraformaldehyde in PBS and permeabilization with 0.5% Triton X-100 in PBS for 5 min, with anti-p62/SQSTM1 (rabbit, Cell Signaling Technology) primary antibodies followed by anti-rabbit Alexa Fluor 488 (A11008, Invitrogen, Waltham, MA, USA). A representative experiment among 3 is shown. The bar graph reports the mean ± SD obtained in three independent experiments.

### 2.7. Apoptosis Evaluation by Flow Cytometry

#### 2.7.1. Propidium Iodide Staining

U87 cells were seeded at a density of 5 × 10^5^ cells/mL per well. After an incubation time of 24 h, 48 h, or 72 h with RDS 3337 at the concentration of 320 nM at 37 °C in 5% CO_2_, cells were collected, centrifuged, and resuspended in a fresh medium. Alternatively, cells were incubated with RDS 3337 for 24 h, 48 h, or 72 h and then with HBSS for 16 h. As a positive control for apoptosis, cells were treated with 1 μM staurosporine (STS) (Sigma-Aldrich) for 8 h at 37 °C in 5% CO_2_. After, they were washed with PBS, fixed in 70% ethanol in PBS for 1 h at 4 °C, washed twice with PBS, resuspended in 125 μL of PBS, 12.5 μL of 5 μg/mL RNase (Sigma-Aldrich), and then stained with 125 μL of 100 μg/mL PI (Sigma-Aldrich). Lastly, cells were incubated for 30 min in the dark at RT before analyzing their DNA content. The fluorescence was measured by a Cytoflex flow cytometer (Beckman Coulter Brea).

#### 2.7.2. AnnexinV/Propidium Iodide (PI)

In U87 cells treated as above, i.e., with 320 nM RDS 3337 for 24 h, 48 h, or 72 h, and then with HBSS for 16 h, they were evaluated by the Annexin V/Propidium Iodide (PI) Apoptosis Detection Kit (556547, BD Biosciences). The different labeling patterns in the annexin V/PI analysis were used to identify the different cell populations: necrotic (annexin−/PI+), late apoptotic (annexin+/PI+), intact (annexin−/PI−), or early apoptotic (annexin+/PI−). The data analysis was performed using a Cytoflex flow cytometer (Beckman Coulter Brea).

### 2.8. Statistical Analysis

All the statistical procedures were performed by GraphPad Prism Software Inc. (San Diego, CA, USA). All data were verified in at least 3 different experiments in duplicate and reported as mean ± standard deviation (SD). Normally distributed variables were summarized using the mean ± SD. The *p*-values for all graphs were generated using Student’s *t*-test as indicated in the figure legends; * *p* < 0.05, ** *p* < 0.005, *** *p* < 0.001, **** *p* < 0.0001.

## 3. Results

### 3.1. HPSE Inhibitor RDS 3337 Promotes LC3-II and p62 Accumulation in RPE-1 Cells

It is known that HPSE has been recently shown to play a role in the autophagy of cancer cells, leading to chemoresistance and tumor progression. Thus, in this study, we first examined HPSE inhibitor RDS 3337 effects on human non-cancer neuro-ectodermal cell line RPE-1, which represents a non-transformed alternative to cancer cell lines. With this aim, we used the HPSE inhibitor RDS 3337, a compound with high anti-HPSE activity, showing nanomolar potency, with an IC_50_ value of 80 nM. First of all, we tested the cytotoxic effect of RDS 3337 under our experimental conditions, using both Trypan Blue and 2-(4-iodophenyl)-3-(4-nitrophenyl)-5-(2,4-disulfophenyl)-2h-tetrazolium, and monosodium salt (WST-1) assays. RPE-1 cells were treated with RDS 3337 at different concentrations (80–1280 nM) and analyzed by both cell counting and cell viability, as shown in Figure 2A,B.

Since the highest concentration (1280 nM) showed a significantly higher cytotoxic effect, as revealed by Trypan Blue, as well as by WST-1 assay, we selected the dose of 320 nM.

Similar findings were found using the BrdU assay (Figure 2C).

Next, to explore the effect of HPSE inhibitor RDS 3337 on the expression of the autophagic marker microtubule-associated protein1 light chain 3 (LC3), we employed immunoblotting analysis in RDS 3337-treated and untreated RPE-1 cells. As shown in Figure 2D, the analysis revealed a mild but significant increase of LC3 II after incubation with 320 nM RDS 3337 for 18 h or 72 h. Quantitative analysis confirmed these data (Figure 2D, see histograms). This finding suggests an accumulation of autophagosomes [34]. As expected, when cells were stimulated with HBSS for 16 h at 37 °C for autophagy induction, the analysis revealed an increase of LC3 II levels together with a significant decrease of p62/SQSTM1 levels, as confirmed by densitometric analysis. This finding indicates the activation of the autophagic flux [34]. On the contrary, when HBSS pre-treated cells were successively exposed to RDS 3337 (18 h or 72 h), the increase of LC3 II was accompanied by a significant enhancement of p62/SQSTM1 levels (Figure 2D, see histograms), indicating an impairment of the autophagic clearance. Indeed, p62/SQSTM1 is a selective autophagy receptor, which sequesters ubiquitinated proteins into autophagosome vesicles by interacting with LC3. Moreover, since p62 is a substrate for autophagic degradation, its degradation can represent a marker of autophagic clearance.

To discriminate between autophagy inducers and blockers, we blocked autophagy with bafilomycin A1 after treatment and evaluated the content of LC3B-II and p62/SQSTM1 by Western blot. Thus, as the control of autophagy flux, RPE-1 cells were treated with BafA1. As expected, cells treated with HBSS and successively with BafA1 displayed a significant accumulation of LC3-II together with a significant increase of p62/SQSTM1 in comparison with control cells (Figure 2D), indicating a block of the autophagic flux [34]. Of interest, also in samples incubated with BafA1 after RDS 3337 treatment for 18 h or 72 h, a significant accumulation of LC3-II, accompanied by a significant increase of p62/SQSTM1, was found (Figure 2D). This finding indicates an effect of the compound in the block of the autophagic flux.

Western blot results obtained, as above, were also confirmed by flow cytometry, using Cyto-ID and anti-p62/SQSTM1 Ab. Following treatment with RDS 3337, the analysis revealed a significant increase in Cyto-ID staining, which indicates autophagosome formation, with a high level of p62/SQSTM1, indicating a blockage of the autophagic flux. These findings confirmed an accumulation of autophagosomes [34]. In parallel, cells stimulated with HBSS for 16 h at 37 °C showed an increase of Cyto-ID staining as compared to control cells, together with a significant decrease of p62/SQSTM1 levels, which indicated the activation of the autophagic flux. On the contrary, in cells pre-treated with HBSS and then with RDS 3337, an increase of Cyto-ID staining without degradation of p62/SQSTM1 was observed, indicating an impairment of the autophagic clearance. As expected, cells pre-treated with HBSS and successively with BafA1 displayed a significant increase of Cyto-ID staining, with a high level of p62/SQSTM1, indicating a block of the autophagic flux. Interestingly, again, in samples incubated with BafA1 after RDS 3337 treatment, a significant increase of Cyto-ID staining, suggestive of autophagosome accumulation, together with a high level of anti-p62/SQSTM1 staining, was found, indicating an effect of the HPSE inhibitor in the block of autophagic flux. In Figure 2E, the mean fluorescence intensities are reported; all the cytofluorimetric panels are shown in Appendix A. As expected, this methodological approach is less sensitive as compared to Western blot for this test.

In sum, these findings indicate that HPSE inhibitor RDS 3337 induces a significant increase of Cyto-ID staining, with a high level of p62/SQSTM1 in RPE-1 cells pre-treated with HBSS, suggestive of an autophagosomes accumulation and a consistent arrest of the autophagic flux.

### 3.2. HPSE Inhibitor RDS 3337 Blocks Autophagic Flux in U87 Human Glioblastoma Cells

Since previous studies have demonstrated that HPSE is highly expressed in many cancer types and is localized within autophagosomes [17], in this current study, we investigated the potential mechanism of the HPSE inhibitor RDS 3337 in regulating autophagy in U87 human glioblastoma cells in which autophagy is markedly increased [35]. For this purpose, we preliminary tested the cytotoxic effect of RDS 3337 in U87 cells, using Trypan Blue (Figure 3A), WST-1 assays (Figure 3B), and BrdU assay (Figure 3C). U87 glioblastoma cells were treated with RDS 3337 at different concentrations (80–1280 nM) and then analyzed either by cell counting or by cell viability, as shown in Figure 3A,B. Similar findings were found using the BrdU assay (Figure 3C). Similarly to RPE-1, we selected the dose of 320 nM also for U87 cell experiments.

Thus, we analyzed the effect of this compound on the autophagic pathway in U87 cells, which basically show quite a high level of LC3-II together with a low level of p62/SQSTM1 (Figure 3D). With this aim, we evaluated the protein levels of the autophagy-related markers, including both LC3-II and p62/SQSTM1, in U87 cells after incubation with 320 nM RDS 3337 for 18 h or 72 h versus control cells (vehicle). As shown in Figure 3D, immunoblotting analysis of lipidated LC3-II and its quantification (histograms on the right) revealed a significant increase of LC3-II in U87 RDS 3337 treated cells compared to untreated ones (vehicle). Interestingly, the analysis of p62/SQSTM1, under the same experimental conditions, showed a significant increase with respect to untreated cells, suggesting a consistent arrest of its degradation. These findings suggest an accumulation of autophagosomes.

Next, to better clarify the action of RDS 3337 in the autophagic process in U87 glioblastoma cells, we verified the autophagic flux using 100 nM BafA1 for 2 h, which prevents lysosomal acidification and, as a consequence, accumulates LC3-II by inhibiting p62/SQSTM1 degradation. Thus, the levels of both lipidated LC3-II and p62/SQSTM1 were assessed by Western blot in RDS 3337-treated cells in combination with BafA1. As shown in Figure 3B, cells treated with RDS 3337 and successively with BafA1 displayed a consistent and significant accumulation of both LC3-II and p62/SQSTM1 as compared to control cells or with RDS 3337-treated cells, indicating a block of the autophagic flux.

All results were also confirmed by using a neuroblastoma cell line, i.e., SK-N-BE2 neuroblastoma cells (Appendix A).

Altogether, these data suggest the ability of HPSE inhibitor RDS3 3337 to induce both LC3-II and p62/SQSTM1 increase, suggestive of an autophagosome accumulation and a consistent arrest in the autophagic flux.

### 3.3. RDS 3337 Inhibitor Sensitizes U87 Human Glioblastoma Cells to Apoptosis

To assess whether the suppression of autophagic flux upon RDS 3337 treatment can activate cytotoxic mechanisms in U87 human glioblastoma cells, we investigated cell death rates following treatment with the HPSE inhibitor. We preliminary determined the activation of caspase 3, a key molecule for induction of caspase and poly (ADP-ribose)-polymerase 1 (Parp1) protein by Western blot analysis in RDS 3337 treated cells. As shown in Figure 4A, RDS 3337 treatment induced a significant increase of the cleaved-caspase 3 in a time-dependent manner, as compared with control cells. Furthermore, we also observed that the level of cleaved Parp1 protein, the nuclear enzyme Parp1 engaged in DNA repair, increased with the time of incubation upon RDS 3337 treatment.

All results were also confirmed by using a neuroblastoma cell line, i.e., SK-N-BE2 neuroblastoma cells (Appendix A).

Data obtained on the proapoptotic activity of RDS 3337 were confirmed by cytofluorimetric analysis using PI staining. These analyses showed a significant time-dependent increase in the sub-G1 phase, which identifies DNA fragmentation, when cells were treated with 320 nM RDS 3337 for 24, 48, or 72 h. Interestingly, after treatment with RDS 3337 for the indicated times (24, 48, or 72 h) and autophagic flux reactivation by HBSS incubation, apoptosis levels were significantly reduced (Figure 4B,D and Appendix A). These findings were also confirmed by annexin V/PI staining (Figure 4C,D and Appendix A), suggesting a balance between autophagy and apoptosis.

## 4. Discussion

Results of the present research support the notion that HPSE promotes autophagy, providing evidence that the RDS 3337 HPSE inhibitor blocks autophagic flux.

HPSE, which traditionally functions extracellularly by cleavage of heparan sulfate and promoting the remodeling of the extracellular matrix (ECM), has long been associated with an increase in tumor metastasis and angiogenesis [36]. However, the recent literature strongly supports a non-enzymatic activity of HPSE in the regulation of intracellular signaling pathways [37]. In this regard, a variety of biological functions and key signaling molecules have been investigated, including cell proliferation, mobility, angiogenesis, and activation of β1 integrin, HIF-2α, Flk-1, and/or AKT signaling [38,39,40,41]. New discoveries have led to the consideration of a possible role for HPSE in modulating autophagy, mainly in the context of tumor growth and chemoresistance, although such a function still remains to be elucidated [12]. A higher increase of autophagy was observed following HPSE overexpression in tumor-derived cells, with an enhancement of tumor growth and chemoresistance. This agrees with a strong pre-clinical and clinical correlation between HPSE expression and the progression of these malignancies [42,43]. Accordingly, studies in HPSE-deficient or transgenic mice established its contributions to autophagy [12].

Enrichment of HPSE in lysosomes suggests that the enzyme may control the normal physiology of these organelles [12,28]. Since it is well known that double-membrane vesicles, called autophagosomes, are fused with lysosomes during autophagy, it cannot be ruled out that lysosomal HPSE may play a role in the completion of autophagy. This emerges from the observation that HPSE is localized within autophagosomes in association with LC3-II in SIHN-013 laryngeal carcinoma cells overexpressing HPSE [12].

More recently, Yang et al. showed that either active or enzymatically inactive HPSE enhanced both autophagosome formation and the expression of related genes in gastric cancer cells [44]. In particular, the presence of non-enzymatic HPSE was able to upregulate both LC3-II protein expression and the level of LAMP2, a lysosomal membrane protein.

In the present study, we analyzed the effect of RDS 3337 HPSE inhibitor on human non-cancer neuro-ectodermal cell line RPE-1 with low levels of autophagy and on U87 human glioblastoma cells, which are endowed with increased levels of autophagy. With this aim, the transit of LC3-II through the autophagic pathway was examined by Western blot analysis. As a rule, autophagic flux is deduced on the basis of LC3-II turnover in both the presence and absence of lysosomal or vacuolar degradation [45]. Here, our findings point out an increase of LC3-II amount both in RPE-1 cells and, more evidently, in U87 human glioblastoma cells following HPSE inhibitor RDS 3337 treatment, suggestive of an autophagosomes accumulation and a consistent arrest in the autophagic flux, as confirmed by cell treatment with BafA1.

Consistently, the RDS 3337 treatment might compromise the pro-autophagy function of the intracellular HPSE, considering the ability of the inhibitor to easily cross the cellular plasma membrane. Up to the present, there are still some important issues unresolved about the role of HPSE inhibitors in affecting autophagic pathways and what are the underlying mechanisms involved. Autophagy related to HPSE appears to involve the mechanistic target of rapamycin complex 1 (MTORC1), an autophagy-suppressive regulator that integrates growth factor, nutrient, and energy signals; inhibition of mTOR1 leads to autophagy induction [46]. Recently, a direct correlation between HPSE overexpression and mTOR1 activity has also been proposed. In fact, in HPSE overexpressing SIHN-013 laryngeal carcinoma cells, mTOR1 activity was downregulated, and this condition positively stimulated the autophagic process [12].

Stress-activated autophagy is known to favor the survival of tumor cells, mostly when apoptosis is defective, protecting them from anticancer therapy, including chemotherapy or radiotherapy, and facilitating multi-drug-resistance development [47]. In this regard, it has been shown that autophagy made MCF-7 cells resistant to apoptosis induced by epirubicin, one of the most effective drugs against breast cancer. On the other hand, inhibition of autophagy by the downregulation of Beclin-1, a crucial upstream protein of autophagy, increased the epirubicin sensitivity of MCF-7 cells by accelerating caspase-9 activity and intrinsic apoptosis [48].

Here, we assessed whether alteration in autophagic activity upon RDS 3337 treatment in U87 human glioblastoma cells could lead to the induction of apoptosis. The relationship between autophagy and apoptosis is complex; similar stimuli can induce either autophagy or apoptosis, and the two phenomena may involve signal transduction pathways, which in turn are dependent on the type of cell nature and duration of stimulus, and stress. In many cases, when the apoptotic response is started, autophagy stops to function, partially because of the caspase-mediated cleavage of essential autophagy proteins [49]; thus, the cascade of caspase-activation associated with apoptosis shuts off the autophagic machinery [50].

Based on these considerations, we investigated the cell death rates by the evaluation of caspase 3 activation. Our results showed that RDS 3337 treatment induced a significant increase, in a time-dependent manner, of the cleaved-caspase 3 as compared with control cells. Furthermore, we also observed that activation of caspase 3 was associated with an increase of both cleaved Parp1 protein, the nuclear enzyme engaged in DNA repair, as well as the sub-G1 phase, which identifies DNA fragmentation of apoptotic cells.

Taken together, the downregulation of the autophagic process may sensitize cells to anticancer drugs, since the deregulation of signal pathways leading to autophagy plays a crucial role in producing critical pathophysiological consequences in numerous cellular mechanisms, including the process of tumorigenesis [51]. Our evidence on apoptosis activation may suggest a role for HPSE inhibitor, making its use particularly advantageous for therapeutic applications, where the progression of tumor growth can be controlled by acting in a balance between apoptosis and autophagy and regulating the autophagic process, without a cytotoxic mechanism affecting cell viability.

## 5. Conclusions

Thus, our findings contribute to further knowledge of autophagy regulation by HPSE, supporting the view that the use of clinically applicable autophagy inhibitors may be one of the important strategies for the control of tumor growth progression.

## Figures and Tables

**Figure 1 cells-12-01891-f001:**
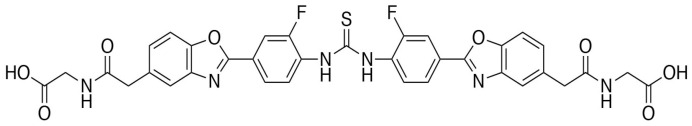
Chemical structure of RDS 3337.

**Figure 2 cells-12-01891-f002:**
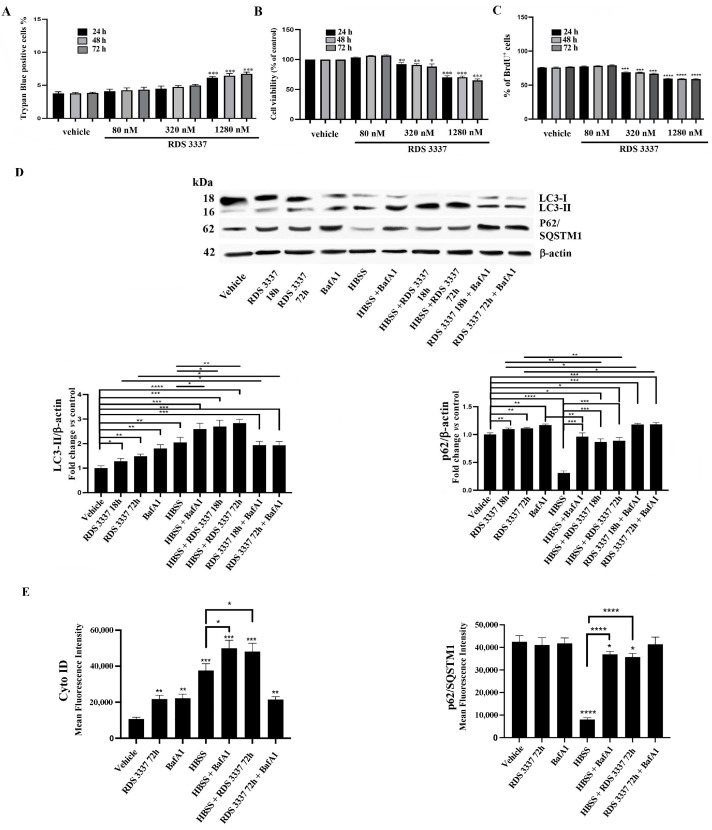
HPSE inhibitor RDS 3337 promotes LC3-II and p62/SQSTM1 accumulation in RPE-1 cells. RPE-1 cells were cultured with 80, 320, or 1280 nM RDS 3337 for 24, 48, or 72 h. (**A**) The number of viable cells was assessed by the Trypan Blue exclusion test. Data are reported as the mean ± SD among five independent experiments. Statistical analysis indicated: *** *p* < 0.0001 versus vehicle. (**B**) Cell viability was measured by 2-(4-iodophenyl)-3-(4-nitrophenyl)-5-(2,4-disulfophenyl)-2h-tetrazolium, monosodium salt (WST-1) assay. Data are reported as the mean ± standard deviation (SD) among five independent experiments. Statistical analysis indicated: * *p* < 0.05, ** *p* < 0.005, *** *p* < 0.001 versus vehicle. We consider cells with only DMSO as vehicle-treated cells. (**C**) Cell proliferation was analyzed by the BrdU incorporation method. Three independent experiments were performed, and a quantitative analysis of the percentage of BrdU^+^ cells is shown. Data are reported as mean ± standard deviation (SD). *** *p* < 0.001 versus vehicle and **** *p* < 0.0001 versus vehicle. (**D**) RPE-1 cells, untreated or treated with 320 nM RDS 3337 for 18 or 72 h, were lysed in lysis buffer. RPE-1 cells were starved with HBSS for 16 h, and the autophagic flux was monitored in the presence or absence of 100 nM Baf A1, or in the presence or absence of RDS 3337 for 18 or 72 h. The samples were analyzed by Western blot, using rabbit anti-LC3 pAb or rabbit anti-SQSTM1 mAb. Anti-actin mAb was used as a loading control. A representative experiment among the three is shown. The bar graph on the right shows densitometric analysis. Results represent the mean ± SD from three independent experiments. * *p* < 0.05, ** *p* < 0.005, *** *p* < 0.001, **** *p* < 0.0001. No statistically significant differences were found between BafA1 and RDS 3337 18 h or 72 h + BafA1 samples. (**E**) Autophagy evaluation by flow cytometry in RPE-1 cells. Cells were starved with HBSS for 16 h, or treated with 320 nM RDS3337 for 72 h, and the autophagic flux was monitored in the presence or absence of 100 nM Baf A1. At the end of treatment, cells were analyzed by flow cytometry after single staining with a Cyto-ID autophagy detection kit. To detect p62/SQSTM1 levels, cells were analyzed by flow cytometry after fixation with 4% paraformaldehyde in PBS and permeabilization by 0.5% Triton X-100 in PBS for 5 min at room temperature, with anti-p62/SQSTM1, followed by anti-rabbit Alexa Fluor 488. The bar graph reports the mean ± SD obtained in three independent experiments. * *p* < 0.05 versus vehicle, ** *p* < 0.005 versus vehicle, *** *p* < 0.0001 versus vehicle, **** *p* < 0.0001 versus vehicle.

**Figure 3 cells-12-01891-f003:**
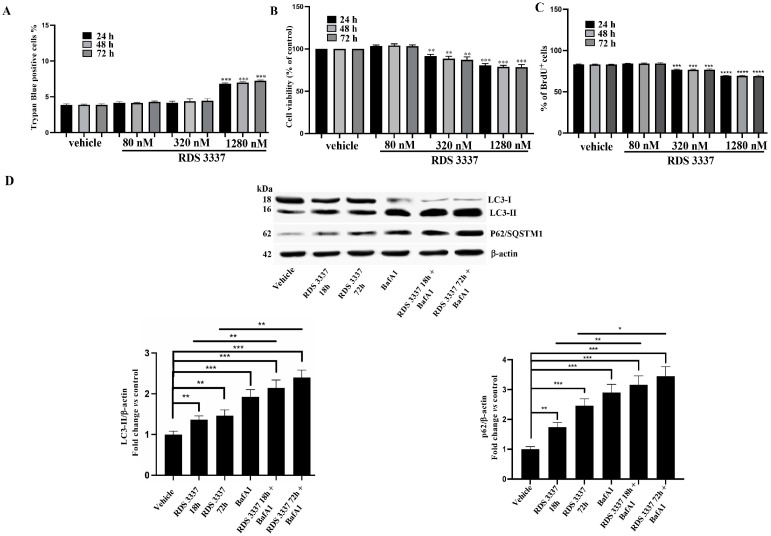
HPSE inhibitor blocks autophagic flux in U87 human glioblastoma cells. Human glioblastoma U87 cells were cultured with 80, 320, or 1280 nM RDS 3337 for 24, 48, or 72 h. (**A**) The number of viable cells was assessed by the Trypan Blue exclusion test. Data are reported as the mean ± SD among five independent experiments. Statistical analysis indicated: *** *p* < 0.001 versus vehicle. (**B**) Cell viability was measured by (WST-1) assay. Data are reported as the mean ± SD among five independent experiments. Statistical analysis indicated: ** *p* < 0.005, *** *p* < 0.0001 versus vehicle. We consider cells with only DMSO as vehicle-treated cells. (**C**) BrdU incorporation analysis. Three independent experiments were performed; percentages of BrdU^+^ cells are shown. Data are reported as mean ± SD. *** *p* < 0.001 versus vehicle; **** *p* < 0.0001 versus vehicle. (**D**) Human glioblastoma U87 cells, untreated or treated with 320 nM RDS 3337 for 18 or 72 h, in the presence or absence of 100 nM Baf A1, were lysed in a lysis buffer. The samples were analyzed for the evaluation of autophagic flux by Western blot, using rabbit anti-LC3 pAb or rabbit anti-SQSTM1 mAb. The loading control was evaluated using anti-actin mAb. A representative experiment among the three is shown. The bar graph on the right shows densitometric analysis. Results represent the mean ± SD from three independent experiments * *p* < 0.05, ** *p* < 0.005, *** *p* < 0.001. No statistically significant differences were found between BafA1 and RDS 3337 18 h or 72 h + BafA1 samples.

**Figure 4 cells-12-01891-f004:**
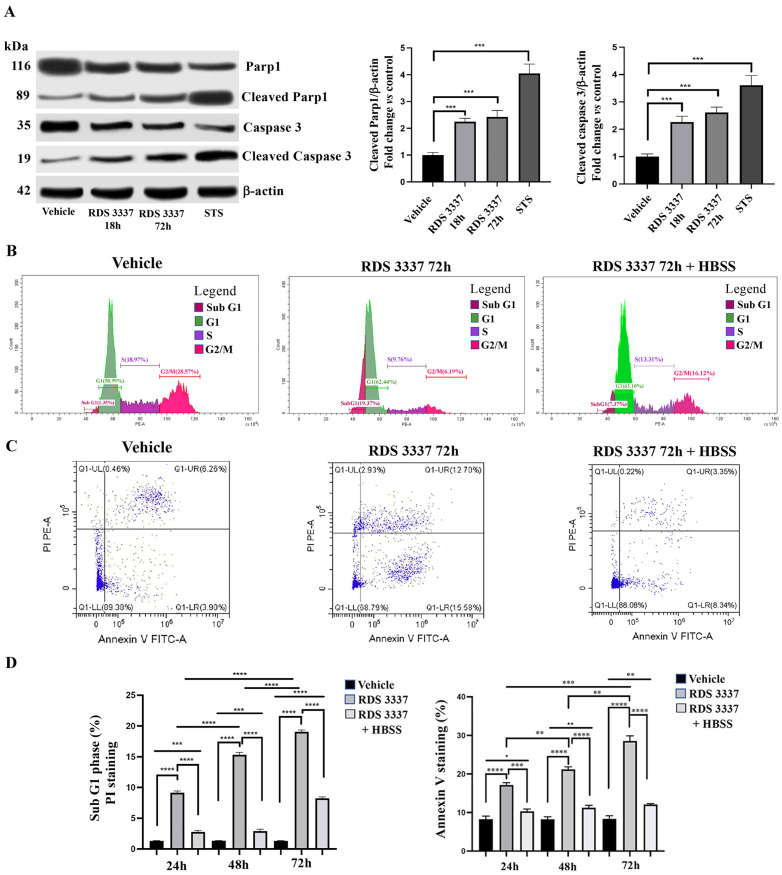
RDS 3337 inhibitor sensitizes U87 human glioblastoma cells to apoptosis. (**A**) U87 cells, untreated or treated with RDS 3337 320 nM for 18 or 72 h, were lysed in lysis buffer. The lysates were analyzed by Western blot to detect caspase three or Parp1 levels, using anti-caspase three mAb or anti-Parp1 mAb. As a positive control, 1 μM staurosporine (STS) was employed. The loading control was evaluated using anti-actin mAb (*left panel*). The bar graph on the right shows densitometric analysis. Results represent the mean ± SD from 3 independent experiments. ** *p* < 0.005, *** *p* < 0.001, **** *p* < 0.0001. (**B**) U87 cells incubated with 320 nM RDS 3337 for 72 h were analyzed by flow cytometric analysis after staining with propidium iodide. Alternatively, cells were incubated with RDS 3337 for 72 h and then with HBSS for 16 h. Representative flow cytometry cell cycle histograms showing subG1, G1, S, and G2/M phases. (**C**) U87 cells, incubated with 320 nM RDS 3337 for 72 h, or with RDS 3337 for 72 h, and then with HBSS for 16 h, were stained with annexin V-FITC/PI before being analyzed by flow cytometry. Representative dot plots of propidium iodide (PI)-Annexin V by flow cytometry are shown. (**D**) The bar graph shows the percentages of the sub-G1 phase that defines apoptosis. Columns and error bars represent the mean ± SD of three separate experiments. *** *p* < 0.001, **** *p* < 0.0001 (*left panel*). On the *right panel*, the bar graph shows the percentages of annexin V-positive cells. The values represent the mean ± SD of three separate experiments. * *p* < 0.05, ** *p* < 0.005, *** *p* < 0.001, **** *p* < 0.0001.

## Data Availability

The data underlying this article will be shared on reasonable request to the corresponding author.

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
