# Peer review of "Role of a Novel Heparanase Inhibitor on the Balance between Apoptosis and Autophagy in U87 Human Glioblastoma Cells"

_cells, 2023, doi:10.3390/cells12141891_

Round 1
Reviewer 1 Report (Previous Reviewer 2)
This round of revision has been completed.
Although the differences with Bafilomycin as conducted in this study are not significant, the rest of the data using e.g. HBSS appear to be significant.
The flux assays could have been further optimized (see not significant effect with Bafilomycin. Eventually, the dosis of BafA1 could have been reduced). The overall quality of the paper has improved during the revisions and also the amount the of the assays of the manuscript has increased, though.
Author Response
We thank the reviewer for his/her comments.
The increased level of p62 in cells treated with BafA1 is more evident in Western blot compared to flow cytometric analysis, since the latter is a less sensitive approach for this test (lines 317-318). Anyway, in this analysis (new Fig. 2E) the high level of p62 is enough to confirm the effect of the compound in the block of the autophagic flux.
Reviewer 2 Report (Previous Reviewer 3)
The authors addressed all my previous questions but I have some final comments:
Fig. 2 and Fig. 3 should be merged as they showed how your drug affects autophagy.
How the authors explained that they didn't see any increase of p62 expression in Baf condition?
Author Response
The authors addressed all my previous questions but I have some final comments:
Fig. 2 and Fig. 3 should be merged as they showed how your drug affects autophagy.
Figure 2 and Figure 3 have been merged.
How the authors explained that they didn't see any increase of p62 expression in Baf condition?
The increased level of p62 in cells treated with BafA1 is more evident in Western blot compared to flow cytometric analysis, since the latter is a less sensitive approach for this test (lines 317-318). Anyway, in this analysis (new Fig. 2E) the high level of p62 is enough to confirm the effect of the compound in the block of the autophagic flux.
Round 2
Reviewer 2 Report (Previous Reviewer 3)
The authors replied to all my comments.
This manuscript is a resubmission of an earlier submission. The following is a list of the peer review reports and author responses from that submission.
Round 1
Reviewer 1 Report
The manuscript by Manganelli V. et al. tries to demonstrate the involvement of Heparanase and its inhibitor RDS3337 on the balance between apoptosis and autophagy in U87 glioblastoma cells.
The authors conducted the experiments more or less correctly (with some exceptions) but strong doubts remain about their capacity to interpret the results. All this leads to conclusions that are incorrect because they are not supported by the results. Furthermore and above all with regard to autophagy which increases thanks to Heparanase these results are already present in a publication by Shteingauz A. et al. (Cancer Res. 75 (18) 3946-57,2015 ).
Major comments:
1) The authors state that the increase of LC3-II and P62/SQSTM1 indicates an arrest of autophagic flux in U87 cells treated with HPSE inhibitor RDS3337. But if the analysis of the autophagic flux by addition of Baf A1 is carefully evaluated, no differences are seen between the control and the cells treated in the presence of the RDS3337 inhibitor. In fact, observing fig 3B, the difference between the BafA1 value and Vehicle is substantially equal to the difference between the RDS3337 (72h)+ BafA1 value and RDS3337 (72h). This indicates that there is no flow variation due to the treatment with RDS3337 and that therefore there is no blockage of the autophagic flow.
2) The authors did not explain how the increased LC3-II accounted for by increased mTOR phosphorylation
3) To demonstrate a cause-effect correlation on apoptosis mediated by the regulation of autophagy modulated by RDS3337 the authors should have also evaluated the effects of an autophagy activation (e.g. rapamycin) in the presence/absence of RD3337 and evaluated the effects on apoptosis.
4) Why was BafA1 not tested in the RPE-1 cell line? This would have allowed to evaluate the arrest of the autophagic flux also in these cells.
Author Response
The manuscript by Manganelli V. et al. tries to demonstrate the involvement of Heparanase and its inhibitor RDS3337 on the balance between apoptosis and autophagy in U87 glioblastoma cells.
The authors conducted the experiments more or less correctly (with some exceptions) but strong doubts remain about their capacity to interpret the results. All this leads to conclusions that are incorrect because they are not supported by the results.
We added several new findings in the new Fig. 2, 5A, 5B and Supplementary Fig. 1, according to the reviewers’ comments. These new data strongly support the conclusions.
Furthermore and above all with regard to autophagy which increases thanks to Heparanase these results are already present in a publication by Shteingauz A. et al. (Cancer Res. 75 (18) 3946-57,2015 ).
Our results confirm but extend the results of Shteingauz, which represent the starting point of our work (Ref. 12 in our manuscript). In particular, in the present study we demonstrate a role for a new HPSE inhibitor on balance between autophagy and apoptosis, indicating that RDS 3337 could act through the inhibition of autophagic-lysosomal flux of LC3-II.
Moreover, the investigated compound RDS 3337 is a novel symmetric derivative of 2-aminophenyl-benzazyl-5-acetate, it is a synthetic small molecule and may offer the opportunity to be adequately designed to achieve favorable pharmacokinetic properties and oral availability, overcoming some of the limitations of polysaccharides that have various limitations, mainly related to their high molecular weight and their heterogeneous nature.
Alternatively, Shteingauz A. et al. described the Heparanase (HPSE) inhibitor PG545, which is structurally different from the one used in our work, in particular, it is a fully sulphated tetrasaccharide functionalized with a cholestanyl aglycone and thus synthesized as a heparan sulphate (HS) mimetic.
Heparanase inhibitor PG545 significantly reduces the survival of U87 cells. Instead, the newly synthesized derivative RDS 3337 did not show antiproliferative activity up to the maximum evaluated concentration (10 μM) [Ref. 31], it was able to inhibit HPSE at concentrations that did not interfere with tumor cell proliferation, making its use particularly advantageous for therapeutic applications, where the progression of tumor growth can be controlled by acting in a balance between apoptosis and autophagy and regulating the autophagic process, without a cytotoxic mechanism affecting cell viability.
Major comments:
1) The authors state that the increase of LC3-II and P62/SQSTM1 indicates an arrest of autophagic flux in U87 cells treated with HPSE inhibitor RDS3337. But if the analysis of the autophagic flux by addition of Baf A1 is carefully evaluated, no differences are seen between the control and the cells treated in the presence of the RDS3337 inhibitor. In fact, observing fig 3B, the difference between the BafA1 value and Vehicle is substantially equal to the difference between the RDS3337 (72h)+ BafA1 value and RDS3337 (72h). This indicates that there is no flow variation due to the treatment with RDS3337 and that therefore there is no blockage of the autophagic flow.
Our results show that LC3-II and p62 levels in BafA1 samples were not significantly different as compared to BafA1 + RDS3337 samples. We added in the legend of Fig. 3B and Supplementary Fig. 2 the statistical analysis. Indeed, as reported by several works, an autophagy inducer results in an increase in LC3B-II, while co-treatment with bafilomycin A1 augments the signal, since the additional autophagosomes that were made are not degraded by the lysosome. In contrast, an autophagy blocker results in an increase in LC3B-II, but the signal is not altered by bafilomycin A1; additional blockage had no influence on the LC3B-II signal, since there was no upregulation of autophagosomes generation.
2) The authors did not explain how the increased LC3-II accounted for by increased mTOR phosphorylation.
This observation is totally in agreement with Shteingauz A. et al. (Cancer Res. 75 (18) 3946-57,2015). However, with the aim to avoid misunderstanding, we decided to remove the results related to mTOR phosphorylation (old Fig. 4).
3) To demonstrate a cause-effect correlation on apoptosis mediated by the regulation of autophagy modulated by RDS3337 the authors should have also evaluated the effects of an autophagy activation (e.g. rapamycin) in the presence/absence of RD3337 and evaluated the effects on apoptosis.
We showed the effect of an autophagy activation induced by HBSS in the presence/absence of RD333. We now show a complete analysis at 24, 48 and 72 h (new Fig. 5A and 5B).
4) Why was BafA1 not tested in the RPE-1 cell line? This would have allowed to evaluate the arrest of the autophagic flux also in these cells.
We already showed the effect of BafA1 in RPE-1 cells. We now added in the new Fig. 2 samples with RDS 3337 + BafA1.
Reviewer 2 Report
The authors have added more data as suggested and the quality of the manuscript has much improved.
One information should be added in the revised version, though:
Figure 3B and Supplement 1: One would like to see the statistical analysis of LC3-II and p62 levels in BafA1 samples vs. BafA1 + RDS3337. Is the increase significant? This should be calculated for evaluation of autophagic flux. Please, add this information.
Author Response
The authors have added more data as suggested and the quality of the manuscript has much improved.
We thank the reviewer for his/her positive comment.
One information should be added in the revised version, though:
Figure 3B and Supplement 1: One would like to see the statistical analysis of LC3-II and p62 levels in BafA1 samples vs. BafA1 + RDS3337. Is the increase significant? This should be calculated for evaluation of autophagic flux. Please, add this information.
Our results show that LC3-II and p62 levels in BafA1 samples were not significantly different as compared to BafA1 + RDS3337 samples. We added in the legend of Fig. 3B and Supplementary Fig. 2 the statistical analysis.
Reviewer 3 Report
In this study, the authors tried to understand how RDS 3337, an inhibitor of HPSE, impacts two different cell lines, U87 and RPE-1. I have several comment for the authors.
1. Authors should described their choice to use RPE-1 and U87 cell lines. using only one cancer cell line is not enough to prove if RDS 3337 may be used as a therapeutic agent in future studies. Authors should also measured expression and activity of HPSE in these cell lines and confirmed that their inhibitor works properly.
2. Most of figures don't have a brief conclusion in the results section, authors should added some.
3. Figure 2:
- In my opinion, trypan blue and WST-1 assay are not convincing methods to measure cell viability. Depending of the count, dilution and time incubation live cells can be coloured by the trypan blue. WST-1 assay measured reduction of WST-1 to formazan. As their inhibitor seems to target mitochondria, authors should confirm that the impact of their inhibitor in this reaction. BrDu or clonogenic assay should be performed to validate the effect of this inhibitor. How the authors explained the difference between data from trypan blue where no significant difference was observed at 320 nM and cell viability?
- Fig 2 and Fig 3 should be merged as they presented same experiments.
- How the authors explained that p62 level is higher in HBSS+RDS compared to HBSS. They should combined Baf and RDS to be sure how RDS impacts autophagy. Moreover, Western blot is not enough to measure autophagy. Authors should add an experiments tracking formation of autophagosome and autolysosome (BacMan RFP-GFP LC3…) with appropriate controls.
4. Fig. 3
- Same comment as Fig. 3
- Why the author did not performed same condition as previously for autophagy measurement. How the authors explained that p62 level is higher in HBSS+RDS compared to HBSS.
5. Fig. 4
- Targets of p-mtor should also be measured
- I disagree about the last sentence (lines 365-366). How authors are sure that decrease of p-mtor is due to inhibition of autophagy by RDS and not the inverse? Induction of p-mtor inhibits autophagy
6. Fig. 5
- Should be merged with Fig. 6 as it showed cell death experiments.
7. Fig. 6
- Authors should also presented the % of double positive cell from flow cytometry and graphs for others conditions
- How the authors explained that there is a difference between time points whereas they didn’t see any difference between them in Fig. 3
- Why the authors did not try to combine HBSS for 24 and 48?
Author Response
In this study, the authors tried to understand how RDS 3337, an inhibitor of HPSE, impacts two different cell lines, U87 and RPE-1. I have several comment for the authors.
- Authors should described their choice to use RPE-1 and U87 cell lines. using only one cancer cell line is not enough to prove if RDS 3337 may be used as a therapeutic agent in future studies. Authors should also measured expression and activity of HPSE in these cell lines and confirmed that their inhibitor works properly.
Two different cancer cells were used in this study, glioblastoma U87 cells and SKNBE2 neuroblastoma cells (see Supplementary Fig. 2). In addition, we used RPE-1 control cells, since these cells represent a non-transformed alternative to cancer cell lines which show low basal level of autophagy.
Activity of HPSE was detected in U87 cells and added in the new Supplementary Fig. 1.
- Most of figures don't have a brief conclusion in the results section, authors should added some.
We added a brief conclusion of each figure in the Results section.
- Figure 2:
- In my opinion, trypan blue and WST-1 assay are not convincing methods to measure cell viability. Depending of the count, dilution and time incubation live cells can be coloured by the trypan blue. WST-1 assay measured reduction of WST-1 to formazan. As their inhibitor seems to target mitochondria, authors should confirm that the impact of their inhibitor in this reaction. BrDu or clonogenic assay should be performed to validate the effect of this inhibitor. How the authors explained the difference between data from trypan blue where no significant difference was observed at 320 nM and cell viability?
WST-1 assay for detection of cell viability was required by a review in previous submission of the manuscript.
This test is more sensitive as compared to Trypan blue assay. It explains why no significant difference was observed at 320 nM and cell viability in the Trypan blue assay.
- Fig 2 and Fig 3 should be merged as they presented same experiments.
We understand the request. However, since we added new data in Figure 2, we feel that merging these figures decrease the quality of the figure. However, we are ready to try to compact the Figures if required.
- How the authors explained that p62 level is higher in HBSS+RDS compared to HBSS. They should combined Baf and RDS to be sure how RDS impacts autophagy. Moreover, Western blot is not enough to measure autophagy. Authors should add an experiments tracking formation of autophagosome and autolysosome (BacMan RFP-GFP LC3…) with appropriate controls.
SQSTM1/p62 is a selective autophagy receptor, which sequesters ubiquitinated proteins into AVs by interacting with LC3. In addition, p62 is a substrate for autophagic degradation, therefore its degradation can be used as a marker of autophagic clearance. One common method to discriminate between autophagy inducers and blockers is to block autophagy by bafilomycin A1 after treatment with the compound under investigation and evaluate the amount of LC3B-II and SQSTM1/p62 by western blot.
We added samples of RDS 3337 + BafA1, 18 h, and RDS 3337 + BafA1, 72 h, in the new Fig. 2B.
- Fig. 3
- Same comment as Fig. 3
- Why the author did not performed same condition as previously for autophagy measurement. How the authors explained that p62 level is higher in HBSS+RDS compared to HBSS ?
We did not perform the same condition since in these cells basal autophagy is quite high.
Thus, in Figure 3 samples treated with HBSS are not shown. Anyway, it is not surprising that p62 level could be higher in HBSS+RDS compared to HBSS since the compound reduces the autophagic flux.
- Fig. 4
- Targets of p-mtor should also be measured
- I disagree about the last sentence (lines 365-366). How authors are sure that decrease of p-mtor is due to inhibition of autophagy by RDS and not the inverse? Induction of p-mtor inhibits autophagy
According to Reviewer 1, we decided to remove the results related to mTOR phosphorylation (old Fig. 4).
- Fig. 5
- Should be merged with Fig. 6 as it showed cell death experiments.
We feel that it is very difficult to merge Figures 5 and 6, since we greatly improved the old Figure 6 with new experiments (new Fig. 5A and 5B).
- Fig. 6
- Authors should also presented the % of double positive cell from flow cytometry and graphs for others conditions
We added all conditions in the flow cytometric panel including RD 3337 treatment for 24 and 48h in the presence of in the absence of HBSS as well as apoptosis analysis by propidium iodide or Annexin V-FITC/PI staining (new Fig. 5A and 5B).
- How the authors explained that there is a difference between time points whereas they didn’t see any difference between them in Fig. 3
The kinetics were different since we studied different phenomena: in the new Figure 5 we analyzed apoptosis, whereas in Figure 3 the autophagic flux.
- Why the authors did not try to combine HBSS for 24 and 48?
We added data at 24 and 48h in the new Figures 5A and 5B.
Round 2
Reviewer 3 Report
Authors replied correctly for most of my comments but did not discuss about important points.
1. If HSPE activity was measured in U87, what about SKNB2? What about cell death in this cell line?
2. Authors did not comment about the possible effect of their compound on activity mitochondria which may affect results of WST-1 assay. Authors must add another experiment to measure directly cell viability OR testing if RDS effect mitochondrial activity.
3. Authors also did not comment about adding an experiment to confirm autophagy inhibition by WB. They must add an experiment to confirm this. Western blot is not enough to confirm autophagy inhibition, even if combinaison of inhibitor and inducer of autophagy were done.
4. Authors said that u87 cell line have a higher autophagy level compared to RP-1. This statement is not show in the paper and must be added. What about SKNB2?
5. Fig. 4 and the new figure 5 should be merged as suggested in my, last review. FACS graph should be moved to supplemental as they are not "necessary" in main figures.
Author Response
Authors replied correctly for most of my comments but did not discuss about important points.
- If HSPE activity was measured in U87, what about SKNB2? What about cell death in this cell line?
We added in the new Supplementary Figure 1 HSPE activity in SKNB2 cells (Fig. 1B).
We added in the new Supplementary Figure 4 the analysis of cell death in these cells showing cleaved Parp1 and cleaved Caspase 3 following treatment with RDS 3337.
- Authors did not comment about the possible effect of their compound on activity mitochondria which may affect results of WST-1 assay. Authors must add another experiment to measure directly cell viability OR testing if RDS effect mitochondrial activity.
To validate the effect of RDS 3337 inhibitor, we added in the new Figures 2 and 3 BrDu assay, as suggested in the last round of review. It indicates that the observed effect is not dependent on mitochondrial activity.
- Authors also did not comment about adding an experiment to confirm autophagy inhibition by WB. They must add an experiment to confirm this. Western blot is not enough to confirm autophagy inhibition, even if combinaison of inhibitor and inducer of autophagy were done.
We added in the new Supplementary Figure 2 the analysis of cell autophagy by flow cytometry, using Cyto-ID Autophagy Detection Kit, which provides a quantitative approach for monitoring autophagic activity at the cellular level by using a 488 nm-excitable probe that becomes fluorescent in autophagosomes produced during autophagy. In the same samples we monitored p62/SQSTM1expression. The analysis showed a significant increase of autophagosomes with high levels of p62/SQSTM1, following RDS 3337 treatment. It is suggestive of an autophagosomes accumulation and a consistent arrest in the autophagic flux.
- Authors said that u87 cell line have a higher autophagy level compared to RP-1. This statement is not show in the paper and must be added. What about SKNB2?
We added the analysis of RPE-1, U87 and SKB2 cells by flow cytometry in the new Supplementary Figure 2. We reported the statistical analysis in the legend.
In addition, comparison of LC3-II bands by Western blot in control cell lines (vehicle in Fig. 2D, 3D and in Supplementary Fig. 3) show higher expression of LC3-II and lower levels of p62/SQSTM1 in U87 glioblastoma and SKNB2 neuroblastoma cells, as compared to RPE-1 cells.
- Fig. 4 and the new figure 5 should be merged as suggested in my, last review. FACS graph should be moved to supplemental as they are not "necessary" in main figures.
Figures 4 and 5 have been merged as suggested. FACS graphs have been moved to the new Supplementary Figures 5 and 6.